# The Response of Rocky Desertification to the Development of Road Networks in Karst Ecologically Fragile Areas

**DOI:** 10.3390/ijerph20043130

**Published:** 2023-02-10

**Authors:** Shiwen Zhang, Yan Wang, Chengrong Li, Yang Wu, Yuhang Yin, Chao Zhang

**Affiliations:** 1College of Ecology and Environment, Southwest Forestry University, Kunming 650224, China; 2Key Laboratory of Ecological Environment Evolution and Pollution Control in Mountainous Rural Areas of Yunnan Province, Kunming 650224, China; 3College of Forestry, Southwest Forestry University, Kunming 650224, China

**Keywords:** road networks, rocky desertification landscape, karst ecologically fragile areas

## Abstract

Frequent cross-regional communication makes road networks increasingly dense and has generated prominent human interference, thus resulting in the destruction of the landscape’s integrity and leading to changes in the functional processes of the habitat. In order to discuss the impacts of intense human activity brought by the road networks on the rocky desertification landscape and habitat quality in karst ecologically fragile areas, taking the road networks as the humans activity intensity factor, a quantitative analysis was conducted to analyze the impacts of road networks on the spatial evolution of the rocky desertification landscape and changes in regional habitat quality characteristics under different development modes in the study area based on a landscape pattern gradient method, spatial analysis, and INVEST model. The results showed that: (1) in the study area, due to the destruction of landscape integrity caused by the development of the road networks over the past 17 years, the landscape pattern of rocky desertification tended to be fragmented and complex, first showing an inclination for rapid fragmentation and then gradual recovery later. (2) The land-use intensity and degree of rocky desertification in the industrial areas and in the tourist areas of the study area have increased to varying degrees over the past 17 years, as is seen mainly via the expansion of construction land, cultivated land enclaves in the urban expansion areas, and new development areas. (3) Unders different regional models, the fragmentation of the rocky desertification landscape in the industrial areas was higher than that in the tourist areas, resulting in a significantly lower habitat quality and obvious degrees of degradation. The research findings provide the basis for further deepening our understanding how human activity intensity affects the evolution of the regional landscape, including the development of rocky desertification, the supply of services, and supporting habitat conservation in karst ecologically fragile areas.

## 1. Introduction

Karst, the general term for karst geomorphic environments and internal geological processes, is an important condition for the development of rocky desertification. Due to the karstification of soluble carbonate rock, the karst geomorphic environment provides an easy avenue for the retrogressive succession of the ecosystem under external interference, leading to the formation of ecologically fragile areas [1] (pp. 16–22). Soluble carbonate rocks exist on the surface of 15.20% of the ice-free continents on Earth and have a population of 1.18 × 10^9^, among which the largest absolute area and population are found in Asia, and the highest percentage is in Europe [2]. The karst rocky desertification area in Southwest China is one of the world’s most concentrated distribution areas of carbonate rocks and the most fully developed continuous karst belt [3], covering an area of more than 5.4 × 10^5^ km^2^ and boasting a carrying capacity of a population of 2.2 × 10^8^ [4]. Karst rocky desertification, an extreme form of land degradation in karst areas, not only encompasses vegetation degradation, soil erosion, bedrock exposure, and ecological environment deterioration in the natural landscape [5], but also encompasses typical ecological fragile areas, ecological security barrier areas, ethnic minority-inhabited areas, and the once largest contiguous poverty area [6]. The governance of the karst rocky desertification area in Southwest China is complicated by the contradiction between ecological security on the one hand, and regional development on the other [7]; thus, the focus is on rural revitalization following poverty alleviation.

Facing the ecological vulnerability and rocky desertification of karst areas, different experts and scholars have developed different understandings and definitions around research and governance. Yuan [8] argues that the ecological fragility of karst areas in Southwest China is aggravated by water and soil loss caused by human activities, which has led to a unique geological environment comprising a weak soil-forming ability and poor soil quality owing to karstification. Tu [9] holds the viewpoint that rocky desertification is a process of land degradation caused by human activity in the karst natural background. Zhou [10] deems that the development of carbonate rock strata and natural vegetation systems has been repeatedly disturbed and destroyed by human activity, which has led to the degradation of the already fragile ecosystem and large-scale exposure of rock strata in various forms, thus forming rocky desertification. Zhang [11] points out that land rocky desertification is the direct result of unreasonable human activity (population pressure and unreasonable economic activity) impacting the fragile geological environment. Wang [12] further expands the category of rocky desertification by arguing that water erosion linked to human activity has resulted in a bare desert landscape of bedrock. Karst rocky desertification is a process of land degradation caused, in part, by the human interference in and destruction of fragile karst environments. To summarize, while the fragile and extremely developed karst geomorphic environment is the natural background and material basis for the formation of rocky desertification, population pressure, unreasonable land use, and human activity are the important drivers. How to better identify the impacts of human activity on the evolution of rocky desertification to reduce the driving force of human activity on the retrogressive succession of karst ecosystems has become a significant scientific problem in the governance process of karst rocky desertification [13,14].

Since the 1990s, due to the processes of urbanization and regional economic development [15], urban development has moved from being loosely associated to agglomerated. The speed and frequency of cross-regional links are important manifestations of regional economic activity, which depends on high-speed and developed road networks [16,17]. With the densification of the road networks, the intensity of human activity along the way increases, and thus, the interference of human activity becomes cumulatively prominent. The significant characteristics of strong development and interference of the regional landscape are touched off, taking the main traffic lines as the development axis to form a cluster area with the rapid development of settlements that are gathered together from the suburbs to the city center [18,19]. The destruction of landscape integrity due to the increasing construction of road networks reduces the natural properties of patches and creates the geographical isolation of some species that depend on natural habitats [20]. At the same time, variations in landscape shape and pattern may also produce a large number of artificial effects at the edge that cause changes in biological and abiotic environments and affect the species richness and the community structure within the patch, thus changing the production capacity and the service function of the habitat. The spatial-temporal variation in the habitat quality is not only the result of the interaction between complex ecological patterns and numerous ecological processes, but is also affected by the environmental matrix of karst ecologically fragile areas. Therefore, clarifying the impacts of the changes in habitat quality in the karst ecosystem caused by the landscape segmentation that is brought about by the development of the road networks is of great practical significance for realizing regional landscape optimization, optimizing habitat quality conservation, and maintaining the stability of the ecosystem functions in the karst ecologically fragile areas.

Taking the road network as the indicator of the human activity intensity, the exploration of the spatial-temporal evolution characteristics and the change rules of the rocky desertification landscape and habitat quality in karst ecologically fragile areas under its influence reveals the response of rocky desertification development and habitat quality change to human drivers brought by road traffic, which will hopefully provide theoretical and technical support for the control of rocky desertification in karst ecologically fragile areas.

## 2. Materials and Methods

### 2.1. Study Area

The study area is located in Xijiekou Town and Changhu Town (24°34′ N~24°53′ N, 103°19′ E~103°40′ E) of Shilin Yi Autonomous County, which is a karst ecologically fragile area in Mid-inner Yunnan province with a total area of 580 km^2^, and is about 65 km^2^ southeast of Kunming City (Figure 1). The terrain of the study area is mainly high in the northeast and low in the southwest. The main external transportation system is distributed in the northern part of the study area along a large number of rural settlements, while only rural roads are scattered throughout the south. Among them, Xijiekou Town is representative of industrial areas, where major roads have been opened successively: Beizhao Road, Yimu Road, and the Shilin section of the Nanning–Kunming Railway. Its land use includes mainly cultivated land, forested land, and grassland. Changhu Town is representative of the tourist areas, where these major roads have been opened successively: the Luquan–Jinma Line, Xiuhe Line, Douyi Road, and Douxi Road, as well as tourist lines directly connected to the Changhu and Shilin Scenic Areas. Its land use includes mainly cultivated land and forested land. The region has a low-latitude plateau mountain monsoon climate, and thus, the annual precipitation distribution is extremely uneven as the precipitation in the rainy season (May–October) accounts for 87% of the annual precipitation, and the spring drought is particularly prominent. The forest quality in the study area is not high, as most of the plant life is sparse shrubs and grasses growing on karst rocky desertification land, resulting in a serious water shortage on the surface and low ecological stability. Because of the poor soil-forming ability of the soluble carbonate rock parent material and thin soil layer, rock desertification easily occurs after extreme natural disasters when coupled with the impacts of unreasonable human activity.

### 2.2. Data Source

Data for 2017 (20 February 2017) were obtained from Google Earth with a spatial resolution of 1 m; data for 2010 (24 December 2010) and 2005 (9 February 2005) were obtained from the Landsat TM with a spatial resolution of 30 m; and data for 2000 (2 November 2000) were obtained from the Landsat ETM+ with a spatial resolution of 15 m. The selected images span from October to April, since this is the dry season with less precipitation and fewer clouds. All the data were pretreated through radiation correction and geometric correction for information extraction, such as to obtain information related to land use and rocky desertification. Socioeconomic data for the past two decades were obtained from *The Statistical Yearbook of Yunnan Province*, the website of digital villages (www.szxc.gov.cn) (accessed on 4 October 2022), and *The Statistical Yearbook of Shilin County*.

### 2.3. Calculation Method

#### 2.3.1. Land-Use Classification

According to *the Classification of Land-use Status* (GB/T21010-2017) issued by the Ministry of Land and Resources in 2017, the land use in the study area can be divided into six types, namely, cultivated land, construction land, forested land, water area, bare land, and grassland, which are given in combination with the distribution of land-use types and research needs.

According to rock outcrop rate (Fr) grading criteria [21], the degree of karst rocky desertification of the land in the study area can be divided into six types, namely, non-rocky desertification (Fr ≤ 0.2), potential rocky desertification (0.2 < Fr ≤ 0.3), light rocky desertification (0.3 < Fr ≤ 0.5), medium rocky desertification (0.5 < Fr ≤ 0.7), severe rocky desertification (0.7 < Fr ≤ 0.9), and extremely severe rocky desertification (Fr > 0.9). Together with the field survey (Table 1) and monitoring data on land use and the degree of rocky desertification in Shilin in 2015 (Figure 2), the patch with the translation error was corrected, and post-treatments after translation, such as patch aggregation and attribute generation, were conducted.

The rock outcrop rate (Fr) was measured based on the Dimidiate Pixel Model with the following equations [21]:
NDRI = (*ρ*_swir2_ − *ρ*_nir_)/(*ρ*_swir2_ + *ρ*_nir_)(1)
where NDRI denotes the normalized differential rocky index, and ρ_swir2_ and ρ_nir_ represent the infrared 2 band and near-infrared band from the Landsat imagery, respectively.
Fr = (NDRI_i_ − NDRI_min_)/(NDRI_max_ − NDRI_min_)(2)
where Fr represents the rock outcrop rate, NDRI_i_ denotes the NDRI value of the ith mixed pixel, NDRI max denotes the maximum NDRI value completely composed of outcrops, and NDRI min denotes the minimum NDRI value completely composed without outcrops.

#### 2.3.2. The Establishment of Network Buffers

The county roads, including Beizhao Road, Yimu Road, the Luquan–Jinma Line, the Xiuhe Line, Douyi Road, Douxi Road, and the railway that go through the two towns of the study area, were set as the central axis. Four buffer radii were selected [22]: 0–1000 m for urban expansion areas; 1000–2000 m for transition areas between urban expansion areas and new urban development areas; 2000–5000 m for new urban development areas; and 5000–9000 m for ecological protection areas (Figure 3).

#### 2.3.3. Landscape Pattern Index

Existing research shows the inadequacy of pattern characteristics used by most landscape indicators, namely, they are sensitive to changes in some individual factors, and slow to respond to other factors. In this study, indices at the landscape level were selected, including the Landscape Shape Index (LSI), Patch Density (PD), Large Patch Index (LPI), Shannon’s Diversity Index (SHDI), Landscape Contagion Index (CONTAG), and Landscape Aggregation Index (AI). At the same time, indices including the Landscape Shape Index (LSI) and Percent of Patches (PLAND) at a class level (different rocky desertification degrees) were also selected (Table 2). A synthetic analysis was conducted at these two levels on different landscape types in buffers of the study area [23,24,25].

#### 2.3.4. Land-Use Intensity

Land-use intensity reflects the degree of human disturbance of land, and it is also the historical accumulation of human disturbance at the level of landscape [26]. According to the characteristics of land-use types located in the karst ecologically fragile study areas, together with previously reported results [27,28], the land-use intensity can be divided into 4 levels, namely: Level 1, the land is hard to use (barren land/land with rock desertification); Level 2, the land is covered with forest, grass, and water (forested land, water); Level 3, the land is for agriculture (cultivated land); and Level 4, the land is for construction (land for residence). The land-use intensity in the study area was calculated through the contribution of land-use types to the land-use intensity as follows:(3)$$Intensityx=\sum_{i=1}^{n}A_{i}\times \frac{S_{i}}{S}$$
where Intensity *x* refers to the comprehensive index of land-use intensity in the *x*th region, *A_i_* refers to the classification index of the land-use intensity at the *i*th level, *S_i_* refers to the land-use area at the *i*th level, and *S* refers to the total land area.

#### 2.3.5. Habitat Quality Model

Habitat quality refers to the ability of the environment to provide suitable conditions for the sustainable survival and development of individuals and populations, which is an important reflection of biodiversity. The quality of habitat determines the sustainable development of environments for human beings alongside nature and other species [29]. High intensity in construction and development will increase the number of threat sources and the intensity of land use, leading to habitat degradation in the adjacent areas. The habitat quality assessment module in the InVEST model takes the land cover as the habitat and assesses the degradation degree of the habitat over a certain period of time due to the threat source from outside the habitat, calculating the habitat quality as follows [30,31]:(1)Calculation of habitat degradation degree *D_xj_*
(4)$$D_{xj}=\sum_{r=1}^{R}\sum_{y=1}^{Y_{r}}\left(w_{r}/\sum_{r=1}^{R}w_{r}\right)r_{y}i_{rxy}\beta_{x}S_{jr}$$
(5)$$i_{rxy}=1-\left(\frac{d_{xy}}{d_{r}\max}\right)\left(\mathrm{Linear}\right)$$
where *D_xj_* refers to the habitat degradation degree, *R* refers to the number of stressors, *Y_r_* refers the grid number of stressors in the graph of land-use types; *W_r_* refers to the weight of the stressor *r*, *r_y_* refers to the number of stressors in each grid in the graph of land-use types, *i_rxy_* refers to the impacts of the stressor *r* in the grid *y* on the grid *x*, and *β_x_* refers to the degree of legal protection. In this study, *β_x_* is not considered; thus, *β_x_* is set as 1. *S_jr_* refers to the sensitivity of land cover in land type *j* to the stressor *r*, *d_xy_* refers to the distance between grid *x* (habitat) and grid *y* (stressor), and *d_rmax_* refers to the impacting scope of stressor *r*.
(6)$$Q_{xj}=H_{j}\left(1-\left(\frac{{D^{z}}_{xj}}{{D^{z}}_{xj}+k^{z}}\right)\right)$$
where *Q_xj_* refers to the habitat quality of grid *x* in land type *j*; *H_j_* refers to the habitat attribute in land-use type *j*; *k* refers to the constant of the semi-saturation coefficient, where k is usually set as 1/2 of the maximum of the habitat degradation degree; and *z* refers to the model default parameter, which is set as 2.5. The habitat quality value ranges between 0 and 1; the larger the value, the more suitable the habitat is.

(2)The establishment of indices

In the InVEST model, land is divided into 2 types: the land with natural attributes, and the land with artificial attributes. The value of ecological stressors (traffic networks) ranges between 0 and 1. According to related findings [31,32] and general principles of biodiversity conservation, in light of the real status of karst landforms in the study area, the attributes of stressors (Table 3) and the sensitivity of habitat types to stressors (Table 4) were determined.

## 3. Results

### 3.1. Influence of Road Networks on the Evolution of the Rocky Desertification Landscape

The landscape pattern of rocky desertification in karst ecologically fragile areas is closely related to the development process [33]. The development of road networks cuts the landscape into fragments; therefore, the interference and impacts of road network development on the evolution of rocky desertification landscape is explored through an analysis of the evolution of the diversity and integrity of the landscape.

#### 3.1.1. Response of Rocky Desertification Landscape Evolution to Overlapping Impacts of Road Networks in Industrial Areas

After 2000, in order to meet the rapid development demands of the northern industrial areas, Beizhao Road, Yimu Road, and the Shilin section of the Nanning–Kunming Railway were opened one after another, laying the foundation for the rapid development of industry in Xijiekou Town. As seen from the landscape composition of rocky desertification patches under the superposition effect due to being different distances from the road networks (Figure 4), the complexity and density of the patch shape first increased and then decreased as the distance from the road networks increased. The rocky desertification landscape with a high density, complex shape, small maximum patch index, and large diversity index in the transition areas between urban expansion areas and new development areas was most affected by the superposition of multiple roads, and thus, the fragmentation degree was the highest of the regions. The rocky desertification landscape in the urban expansion areas with simple shapes, a low patch density, and less diversity tended to have no fragmentation.

From the perspective of spatial-temporal changes, the total number and density of patches in the study area increased, and the complexity of the shape gradually increased over the course of the past 17 years. From 2000 to 2005, a great quantity of light and medium rocky desertification patches gradually changed to severe/extremely severe rocky desertification owing to the complexity of landscape shape. This is because the dominant patches with medium/light rocky desertification or simple shapes in the transition areas between urban expansion areas and new development areas shifted to severe/extremely severe levels with complex and broken shapes, leading to the decline in the landscape spread degree and the increase in the landscape fragmentation degree in the regions. In the new development areas and ecological protection areas, the dominant patches were light rocky desertification patches with the lowest landscape fragmentation degree and the highest landscape spread degree. Only when the edges had a complex shape were the rocky desertification patches transferred to the level of severe rocky desertification. Since the shape of rocky desertification landscapes tended to be simple and the degree of fragmentation decreased from 2005 to 2017, the medium/severe rocky desertification patches in the transition areas between urban expansion areas and new development areas started to decrease, and the degree of rocky desertification in the new development areas and ecological protection areas shifted to the degree of light/potential rocky desertification.

#### 3.1.2. Response of Rocky Desertification Landscape Evolution to Overlapping Impacts of Road Networks in Tourist Areas

Since 2005, the tourist areas have vigorously developed tourism around the Changhu Scenic Area in Shilin County, successively opening special tourism lines and the Douyi–Douxi Highway to gradually begin to transform industry and agriculture into ecotourism. As seen from the landscape composition of rocky desertification patches under the superposition effect due to being different distances from the road networks (Figure 5), the complexity and density of the patch shape first increased and then decreased. The rocky desertification landscape with a high density, complex shape, small maximum patch index, and large diversity index in the transition areas between urban expansion areas and new development areas was most affected by the superposition of multiple roads, and thus, the fragmentation degree was the highest of the regions. The rocky desertification patches in the urban expansion areas and ecological protection areas with a low landscape density, simple shape, good landscape spread, and high aggregation index tended to be integrated with no fragmentation.

From the perspective of spatial-temporal changes, the medium/severe rocky desertification patches had complex shapes and accounted for a large proportion of landscapes from 2000 to 2005, which was due to the fact that the light/potential rocky desertification patches, which accounted for a large proportion in urban expansion areas including new development areas in the transition areas, were rapidly degrading to the severe/extremely severe rocky desertification level because of the broken edges caused by the construction activity of reclamation, farming, and tourism infrastructure. From 2005 to 2017, the launch of ecotourism projects accelerated the ecological governance measures in these regions, performing a function in curbing the occurrence and development of rocky desertification. In the ecological protection areas, due to traffic occlusion and the lack of necessary human ecological governance, abandoned farmland that was reclaimed in the early stage turned into bare land or wasteland in the later stage because of the slow ecological recovery. The loss of covering soil in this part of the patch led to rocky desertification, owing to the lack of vegetation coverage.

### 3.2. Influence of Road Networks on Rocky Desertification Ecosystem

#### 3.2.1. Change of Land-Use Intensity in the Karst Areas under the Development of Road Networks

According to the changes of land-use intensity [34,35] in different buffers around the road networks of the study area over the past 17 years (Figure 6), the overall trend in the industrial areas centered around the axis of human activity in the main road networks, which decreased slightly in concentric bands from the center to the periphery with spatial values from 1.72 to 2.32. Among them, the land-use intensity was the highest in the urban expansion areas, and was the lowest in the transition areas. The land-use intensity changed the least during 2000–2005, increased significantly during 2005–2010 in all regions, and decreased significantly during 2010–2017. The overall trend of the value of land-use intensity in the tourist areas over the past 17 years: The land-use intensity was the highest in urban expansion areas and the lowest in ecological protection areas. The land-use intensity decreased gradually in the urban expansion areas, new development areas, and transition areas during 2000–2005, and increased generally during 2005–2010. During 2010–2017, the land-use intensity in the transition areas reached the highest level, and then showed a continuous decreasing trend, whereas the land-use intensity was the lowest in the ecological protection areas.

From Figure 7, we can clearly see the distribution proportion and overall trend of the land-use landscape from the five buffer gradients. The areas with serious rocky desertification in the industrial areas were concentrated far away from the traffic centers. Centralized farming was more scientific than the traditional “local and nearby” retail farming, as it caused less damage and burden to the land. The residential land in the urban expansion areas had the largest ratio, and the cultivated land in the regions was largely outsourced; therefore there was a potential/light degree of rocky desertification. The main source of farming was ordinary farmers who have reclaimed the wasteland in the transitional areas between the urban expansion areas and the new development areas for planting mainly traditional cash crops such as corn and flue-cured tobacco. Thus, unreasonable human activity such as farming methods, reclamation methods, and deforestation has caused the rapid development of rocky desertification in the regions. Since roads were opened one after another and industrial areas developed rapidly during 2000–2010, the rocky desertification landscape was concentrated in new development areas and ecological protection areas. From 2010 to 2017, due to the planting of economic forests, such as apples and walnuts in the Stone Forest Areas, and the implementation of control measures for rocky desertification in the ecological protection areas, some potential/light rocky desertification land was converted into forested land and cultivated land, resulting in the increase in forested land and cultivated land area, as well as the decrease in rocky desertification.

From 2000 to 2017, the cultivated land in the tourist areas showed a trend of first increasing and then continuously decreasing. The area of rocky desertification decreased while the area of forested land increased significantly. In regard to the trend of cultivated land in different buffers of the road networks: While the cultivated land areas in the new development areas continued to rise, the ecological protection areas significantly decreased in the latter period; the forested landscape areas were the smallest in the transitional areas between urban expansion areas and new development areas, and the largest in the ecological protection areas. Since the protection policy was implemented in the 1000 m lake buffer around the Changhu Lake in 2006, and despite a large flow of people to the regions, the Shilin government’s lake management policy for the scenic area and Changhu Lake helped to avoid many adverse farming and land reclamation activities. Therefore, a large amount of medium/severe rocky desertification has changed into potential/light rocky desertification in the ecological protection areas due to the significant increase in forested land, the decrease in farming areas, and the significant decrease in the rocky desertification level during 2005–2017.

#### 3.2.2. Analysis of Habitat Quality Evolution in the Karst Areas under the Threat of Road Networks

Not only can the habitat quality index reflect the fragmentation of the habitat patches in the study area, but it can also reflect the ability of the habitat patches to resist the threat of habitat degradation caused by the development of road networks. The continuous value is between 0 and 1. The closer to 1, the better the habitat quality is; the closer to 0, the worse it is. The calculated results of the habitat quality index in the study area are shown in Figure 8.

Combined with land use (Figure 1), the habitat quality of Changhu Town in the southern part of the study area was obviously better than that of Xijiekou Town in the north from the perspective of spatial scale. The main reason was that the number and area of forested land patches in the south were significantly more than those in the north, whereas the cultivated land and grassland in the north occupied a larger proportion of the area. From the view of patch properties, the habitat attributes of forested land patches were obviously better than those of cultivated land and grassland, so that the habitat quality of the southern part was better than that of the northern part. According to the rock desertification level (Figure 2), the rock desertification level of Changhu Town was obviously lower than that of Xijiekou Town, whereas the habitat quality of patches with less rock desertification showed better uniformity. The overall result was consistent with the judgment that the rocky desertification level was negatively correlated with the habitat quality. However, by comparing the rocky desertification level and habitat quality in different years, it could be found that in 2005, when the rocky desertification degree was relatively moderate, the habitat quality was in poor form. Thus, the influencing factors of habitat quality were not only due to rocky desertification, of which the mechanism should be further discussed in subsequent research.

In terms of time scale, the habitat quality in the study area declined significantly from 2000 to 2017, and especially in 2005, during which the rocky desertification land was developed strongly, and the large reduction in forested land and grassland caused severe damage to the habitat quality. From 2010 to 2017, the areas of patches with good and superior habitats continued to increase, in which superior patches were largely concentrated in the tourist areas. Based on the development orientation of ecotourism, the tourist areas have successively carried out a series of ecological protection and rocky desertification restoration measures such as returning farmland to forests and building scenic reserves, which has promoted the transfer of low- and middle-grade habitat patches to be high-grade patches around the scenic area. The industrial areas were mainly composed of low-grade patches that were concentrated in the urban expansion areas and the transition areas near the northern industrial areas. The opening of the Nanning–Kunming Railway has accelerated the development of stone mining and industry, resulting in the restriction of regional ecological industry, which has significantly reduced the quality of the habitat in those regions.

#### 3.2.3. Analysis of Habitat Quality Changes in the Karst Areas under Different Industrial Structures

In order to more intuitively reflect the spatial-temporal change characteristics of habitat quality degradation under different industrial structures in the study area, the results from 2000 to 2017 were calculated by using the Raster Calculator tool from the ArcGIS10.2 (Figure 9).

The overall habitat quality of the tourist areas has declined. In 2005, due to the substantial increase in area of cultivated land, a large number of land reclamation and farming activities led to rocky desertification in the lands around the cultivated land, as well as the intensity upgrade of rocky desertification in the central part. Due to the implementation of the government’s Changhu protection policy in 2010, the intensity of some light/medium rocky desertification lands in Douhei and Suoyishan near the center of Changhu Town began to decline as a large number of forests and grasslands emerged; furthermore, Dazolong, which is near the Changhu Scenic Area, avoided a large amount of unreasonable reclamation in this area. By 2017, with the strong support of the state, the opening of tourist lines and high-speed lines have led to the transformation of the emerging tourist areas, moving of the economic and industrial structure from a single industry to a model of joint development via planting, tourism services, and stone processing, whereas ecological tourism has become the main source of income for towns and villages after planting. With economic development, Shilin County began to implement the policy of controlling and restoring rocky desertification in a large area in 2013, and thus, the degradation of the habitat quality in this area has been significantly alleviated.

The backward mode of production and low management level of enterprises in the industrial areas was mainly due to their income from the planting industry, as well as the stone mining and processing industry. As the roads were gradually put into use, large quarries were opened in Nuoyi Village in the southwest of Xijiekou in 2005. Open-pit mining led to strips of massive topsoil causing severe damage to the landform landscape and land cover during the production process, resulting in the desertification of land with bare rocks, barren land, and scattered stones. In 2010, the construction of solar power plants in the north-central part of the region brought about the land-classification transition from many light/medium rocky desertification lands to man-made lands. Thus, the habitat quality was seriously degraded, and the rocky desertification of the surrounding land was gradually increased. During the past 17 years, although the area of severe rocky desertification in the regions has increased significantly, an extremely severe rocky desertification landscape has appeared in the center, leading to a significant decline in habitat quality. The population aggregation caused by the dense road networks and economic development demand have had dramatic impacts on the change in the regional landscape structure, increasing the threats to the habitat and making the degradation of the habitat quality in the industrial areas significantly higher than that in the tourist areas.

## 4. Discussion

### 4.1. Response of Rocky Desertification Landscape Pattern to Road Network Development

According to the development characteristics of rocky desertification in the study area from 2000 to 2017, the highly intensive rocky desertification landscape with scattered distribution, fine patches, and an irregular shape was embedded into the low-degree rocky desertification landscape. The landscape pattern in urban expansion areas and ecological protection areas had better integrity with the slow development of rocky desertification. The urban expansion areas were close to the residential areas, and the farming activity was mainly concentrated on the contracted land [36]. Due to the low accessibility of human activity and the implementation of governing policies, the land in the ecological protection areas was less disturbed. The areas with the strong development of rocky desertification spread outward from the transition areas between urban expansion areas and new development areas to the new development areas, which were most affected by the superposition of multiple roads [37]. With the reduction in land contracting, the increase in farmers’ individual production activity led to the dispersion of cultivated land and the increase in wasteland. Together with its special geological structure, the land was prone to serious water and soil loss and rocky desertification, which led to the reduction in cultivated land areas that could meet the original farming requirements. The farmers have shifted their high-intensity farming practices to potential rocky desertification areas, but these areas are far away [38]. Intense and unreasonable human activity caused serious land degradation, which in turn, affected regional economic development and the intensity of human activity [39,40]. Again and again, the distribution of land-use types was restricted by the development of fragmented rocky desertification land, causing the fragmentation of regional landscapes. The expansion and transformation of rocky desertification-type land was far more complex than the transformation among simple types of land use, for which the expansion and development was not just a simple area expansion, but also a transformation between intensity types [41]. Eventually, the area or intensity of rocky desertification increased across the region as a whole, resulting in the development differences in the rocky desertification landscape in karst ecologically fragile areas.

### 4.2. Response of Habitat Quality Evolution to Road Network Development

From 2000 to 2017, from the perspective of habitat quality assessment under the superposition effect of road networks that are located at different distances in the study area, the areas with low habitat quality were generally distributed in urban expansion areas and transition areas. The above two areas were most threatened by the superposition of multiple roads in the case that the main roads were the threat sources. The urban expansion areas had the strongest intensity of land use, most of which were residential areas, and thus the suitability of the basic habitat was low. In the transition areas, the land-use intensity was the smallest for scattered farmland in large areas. Long-term cultivation led to the development of rocky desertification and a reduction in land productivity and utilization efficiency [42], which restricted the evolution of land-use types in the regions due to the development of fragmented and distributed rocky desertification landscapes, and brought about a low level of habitat quality in the regions. The areas with a significant decline in habitat quality were concentrated in the transition areas and new development areas. The wasteland reclamation, infrastructure construction, and other industrial activities in the regions have made the dominant land use be classified as cultivated land/garden land, wasteland, and rocky desertification land. Under long-term human interference, these three types of land with a low suitability for basic habitats and a high ecological sensitivity can easily result in the serous degeneration of habitat quality for land degradation [43,44].

### 4.3. Differential Influence of Regional Models on Ecological Environment

Due to the different development models of karst ecologically fragile areas in mid-inner Yunnan province, the intensity of impacts on the regional ecological environment was also different. From 2000 to 2017, the development degree of rocky desertification and changes in habitat quality in buffers at the same distance from the road networks were different under different development models in the same regions of the study area. On the spatial-temporal scale, the area of rocky desertification have increased in tourist areas with better economic development in the south of the study area, but the landscape with a relatively high level of habitat quality tended to be complete with no fragmentation and an inapparent degree of degradation under the influence of multiple roads [45]. This phenomenon of rocky desertification was due to the significant increase in the area and the significant aggravation degree in the study area from 2000 to 2005, which has gradually become an obstacle to the development of agriculture and forestry in Shilin County. The tourist areas gradually shifted the focus of regional development from planting and the processing industry to tourism, and thus, the areas of abandoned land and economic forests have increased. Moreover, artificial intervention to control rocky desertification was carried out around the scenic areas. In addition, the regional landscape pattern was relatively complete and the ecosystem had a strong ability to recover [46]; therefore, the change in the regional model had positive impacts on its ecological environment. The rock desertification degree increased as the landscape pattern became more fragmented and complex, and as the industrial areas with a high intensity and significant degradation degrees were at a low level of habitat quality. This was because the areas were abundant in stone resources and far away from the Stone Forest Scenic Area. Later, the opening of the Nanning–Kunming Railway made stone mining, processing, and transportation become an industry with high economic benefits for the region. In addition, farming and animal husbandry were the main economic sources for rural farmers due to the flatness of the region, and thus, the ecological environment was strongly affected by unreasonable human activity with significant negative impacts [47].

## 5. Conclusions

The karst ecologically fragile areas in mid-inner Yunnan province (Shilin Yi Autonomous County, Yunnan) were taken as the research object to calculate the impacts of road network development on the habitat quality in the karst areas under two different development modes of industry and tourism by using the InVEST model, to analyze the spatial-temporal evolution characteristics with the help of GIS tools, and to analyze the evolution law of rocky desertification in the karst areas with the use of the landscape pattern index. The conclusions are as follows:(1)From 2000 to 2017, the landscape pattern of rocky desertification in the study area first showed a development trend of rapid fragmentation and then gradual recovery later owing to the destroyed landscape integrity, significantly decreased ecological value, and the rapidly increasing areas of rocky desertification patches. The development of rocky desertification in different buffers was obviously different, in which the rocky desertification patches in urban expansion areas, new development areas, and transition areas were most affected by the superposition of multiple roads; thus, the landscape patterns were evolving towards having fragmentation, complexity, and heterogeneity.(2)From 2000 to 2017, the intensity of land use and the degree of rocky desertification in the industrial areas and tourist areas increased to varying degrees, mainly in the form of construction land and farming land enclaves that were expanded in the urban expansion areas and the new development areas. Over the course of the rapid development of the industrial areas, ecological protection areas have been partially occupied by the significant increase in cultivated land. The infrastructure construction of the tourist areas in the early period has led to the increase in land-use intensity in new development areas and transition areas. Later, the implementation of ecological governance measures has increased the area of forested land, and thus, the problem of rocky desertification in the regions has been significantly improved.(3)Different regional development patterns led to different impacts of road network development on rocky desertification development and habitat quality changes in buffer areas. The landscape pattern of rocky desertification in the industrial areas, of which the fragmentation degree was higher than that of tourist areas, was affected by the change in land use with a significant degree of degradation, resulting in a significantly lower habitat quality than that of tourist areas.

The study focuses on the influence of human activities on rocky desertification landscapes and habitat quality in karst ecologically fragile areas under different regional models. However, human activities do not only include those associated with road networks. In a follow-up study, the response of the rocky desertification landscape in karst ecologically fragile areas to human activities can be explored from the aspects of population density, economic development, settlement expansion, tourism development, etc. The influence mechanism can be comprehensively interpreted to expand the scope and depth of this study. For different types of ecologically fragile areas in the world, human activities are inevitable influencing factors, and thus, the study of this topic is of great significance for the construction of ecological restoration projects and the exploration of formation mechanisms.

## Figures and Tables

**Figure 1 ijerph-20-03130-f001:**
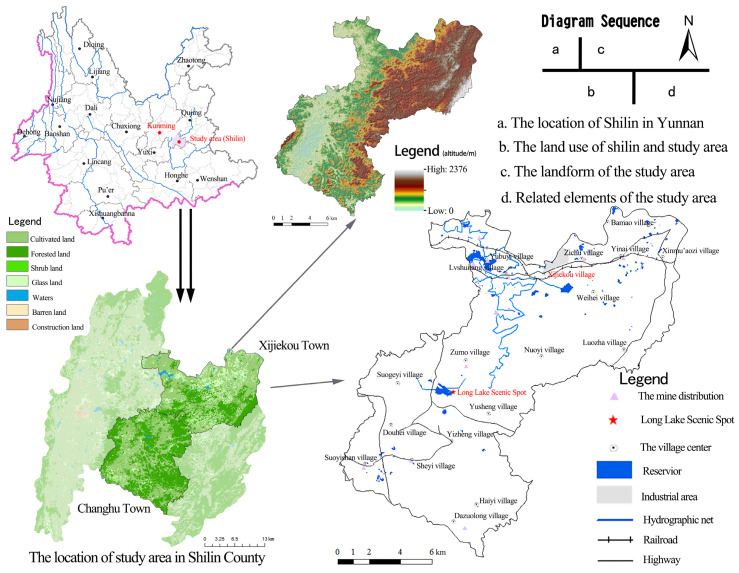
Overview of the study area.

**Figure 2 ijerph-20-03130-f002:**
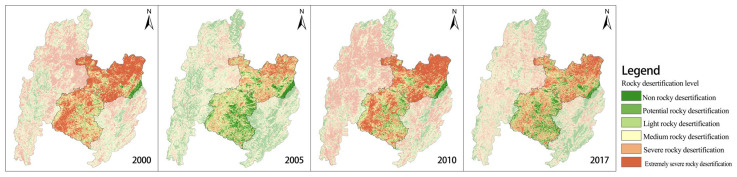
Dynamic evolution of rocky desertification level in the study area from 2000 to 2017.

**Figure 3 ijerph-20-03130-f003:**
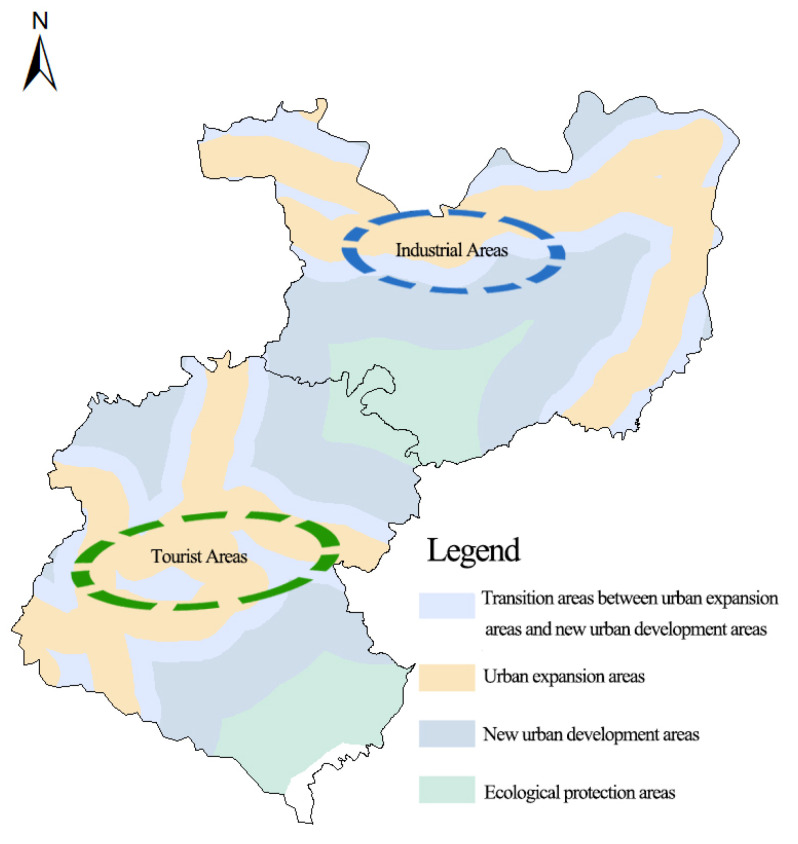
**The** buffer of road networks.

**Figure 4 ijerph-20-03130-f004:**
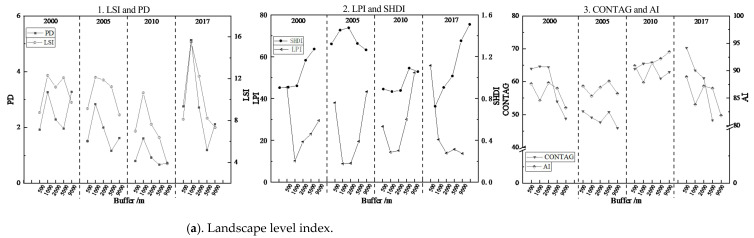
Spatial evolution of rocky desertification patches with different buffers in the industrial areas from 2000 to 2017. **Note:** a—non-rocky desertification; b—potential rocky desertification; c—light rocky desertification; d—medium rocky desertification; e—severe rocky desertification; f—extremely severe rocky desertification.

**Figure 5 ijerph-20-03130-f005:**
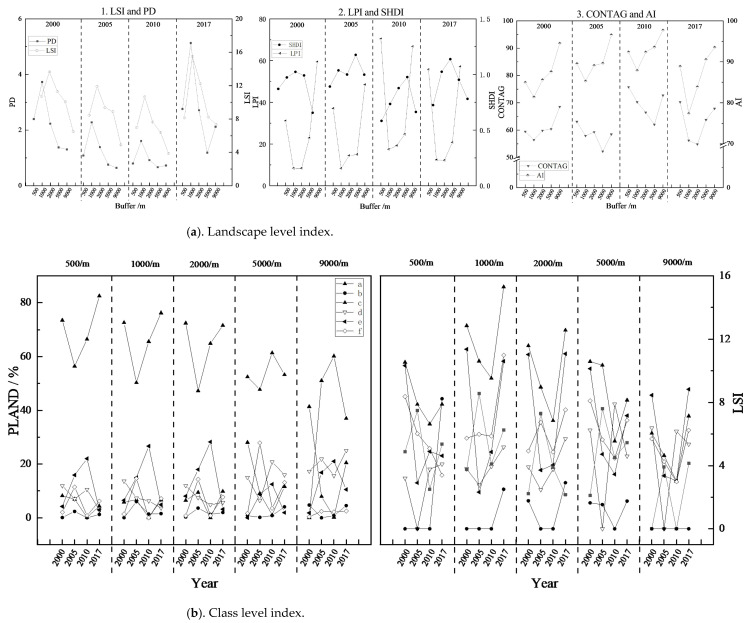
Spatial evolution of rocky desertification patches with different buffers in the tourist areas from 2000 to 2017. **Note:** a—non-rocky desertification; b—potential rocky desertification; c—light rocky desertification; d—medium rocky desertification; e—severe rocky desertification; f—extremely severe rocky desertification.

**Figure 6 ijerph-20-03130-f006:**
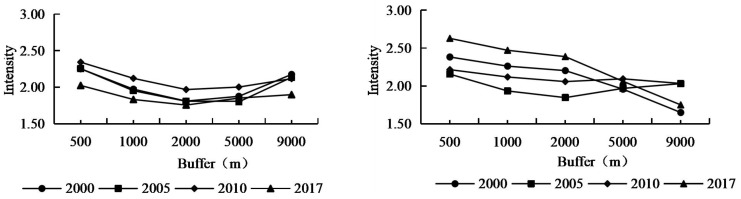
Land-use intensity of different buffers in the study area from 2000 to 2017.

**Figure 7 ijerph-20-03130-f007:**
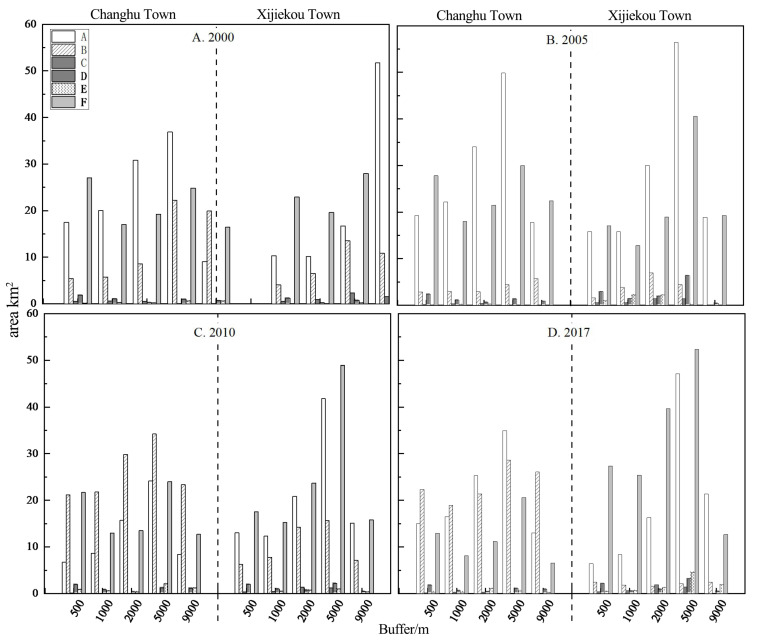
Land-use landscape composition of different buffers in the study area from 2000 to 2017. **Note:** A—non-rocky desertification; B—potential rocky desertification; C—light rocky desertification; D—medium rocky desertification; E—severe rocky desertification; F—extremely severe rocky desertification.

**Figure 8 ijerph-20-03130-f008:**
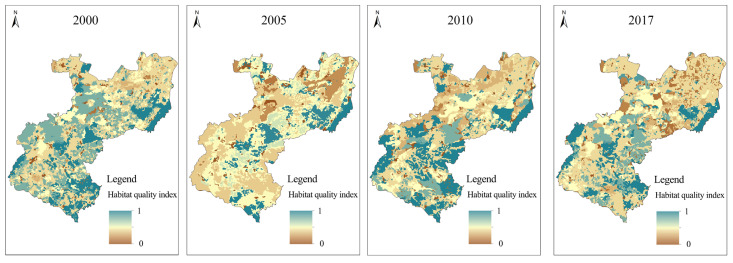
Habitat quality change in the study area from 2000 to 2017.

**Figure 9 ijerph-20-03130-f009:**
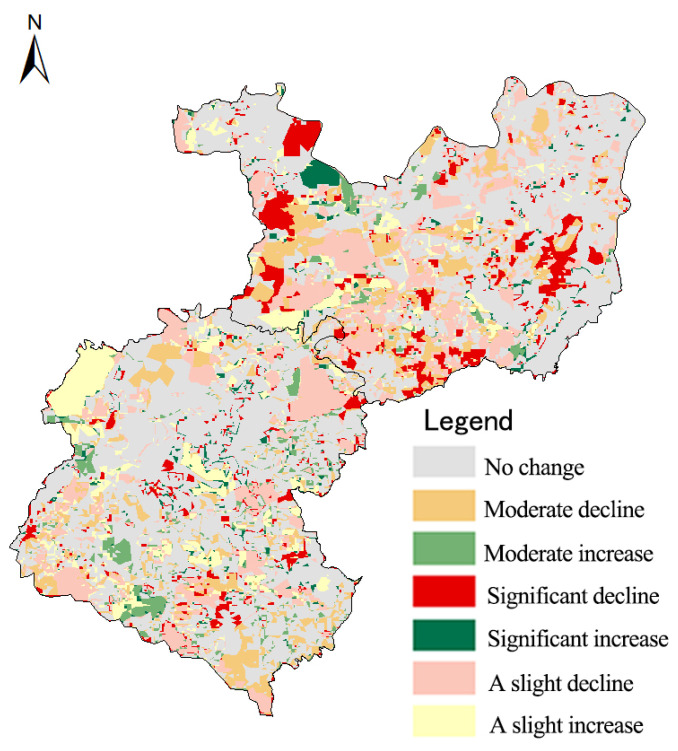
The degree of habitat quality degradation in the study area from 2000 to 2017.

**Table 1 ijerph-20-03130-t001:** Actual investigation of the rocky desertification degree in the study area.

Altitude/m	Classification	Actual Investigation	Satellite Comparison
1957	Non-/potential rocky desertification (Fr ≤ 0.3)	** 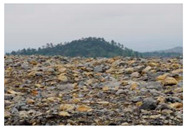 **	** 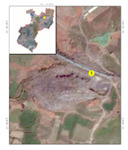 **
2165	Light rocky desertification (0.3 < Fr ≤ 0.5)	** 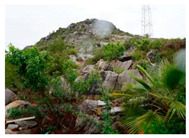 **	** 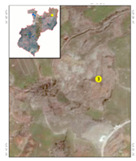 **
2155	Medium rocky desertification (0.5 < Fr ≤ 0.7)	** 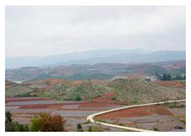 **	** 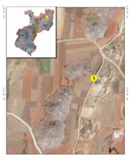 **
1907	Severe rocky desertification (0.7 < Fr ≤ 0.9)	** 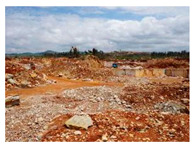 **	** 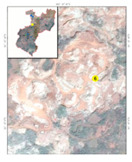 **
1800	Extremely severe rocky desertification (Fr > 0.9)	** 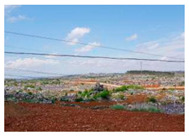 **	** 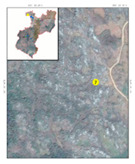 **

**Table 2 ijerph-20-03130-t002:** Landscape pattern index and basic information.

Type	Index	Computing Method	Ecological Significance
Landscape Level	Landscape Shape Index	$\mathrm{LSI}=\frac{E}{\min\;(E)}$	The complexity of patch shape is determined by calculating the deviation between a patch shape and a circle or square with the same area.
	Patch Density	PD=NP/A	The fragmentation degree of landscape type is reflected by the number of patches per unit area.
	Large Patch Index: %	$\mathrm{LPI}=\frac{{\max(a}_{\mathrm{ij}})}{A}(100)$	The proportion of the largest patch area aij in the landscape to determine the dominant patch type.
	Shannon’s Diversity Index	$\mathrm{SHDI}=-\sum_{i=1}^{m}\left(P_{i}\times \mathrm{In}P_{i}\right)$	As the measure of landscape heterogeneity and diversity, the richer the patch type is, the higher the uncertainty and the higher the degree of fragmentation is.
	Landscape Contagion Index: %	$\mathrm{CONTAG}=\left[1+\frac{\sum_{i=1}^{m}\sum_{k=1}^{m}\left[\left(P_{i}\right)\left\{\frac{{\text{g}}_{ik}}{\sum_{k=1}^{m}{\text{g}}_{k}}\right\}\right]•\left[ln\left(Pi\right)\left\{\frac{{\text{g}}_{k}}{\sum_{k=1}^{m}{\text{g}}_{k}}\right\}\right]}{2ln\left(m\right)}\right](100)$	It reflects the degree of aggregation or extension trend of different patch types. With the increase in the index, the better the connectivity between the dominant patches is, and the higher the fragmentation is.
	Landscape Aggregation Index: %	$\mathrm{AI}=\left[\sum_{i=1}^{m}\left(\frac{g_{ii}}{\max g_{ii}}\right)P_{i}\right]\left(100\right)$	It reflects the aggregation degree of landscape types in space. With the increase in indicators, the aggregation degree of patches increases.
Class Level	Landscape Shape Index	$\mathrm{LSI}=\frac{E}{\min\;(E)}$	The complexity of patch shape is determined by calculating the deviation between a patch shape and a circle or square with the same area.
	Percent of Patches: %	$\mathrm{PLAND}=\frac{\sum_{j=1}^{n}a_{ij}}{A}(100)$	It represents the relative proportion of a patch type in the whole landscape area to identify the dominant patch.

**Table 3 ijerph-20-03130-t003:** Attributes of stressors.

Stressors	Maximum Impacting Distance	Relative Weight Value	Spatial Recession Type
Highway	8	0.6	linear
Railway	2	0.8	linear

**Table 4 ijerph-20-03130-t004:** Sensitivity of habitat types to stressors.

No. of Land-Use Types	Name of Land-Use Type		Habitat Suitability	Highway	Railway
1	Cultivated land		0.4	0.4	0.3
2	Construction land		0	0.8	0.75
3	Forested land		1	0.8	0.75
4	Barren land	Potential rocky desertification	0.1	0.8	0.6
5	Water		0.9	0.2	0.15
6	Grassland	6-1 Light rocky desertification	0.8	0.8	0.75
		6-2 Medium rocky desertification	0.5		
		6-3 Severe rocky desertification	0.3		
		6-4 Extremely severe rocky desertification	0.1		

## Data Availability

Not applicable.

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
