# Peer review of "The Response of Rocky Desertification to the Development of Road Networks in Karst Ecologically Fragile Areas"

_ijerph, 2023, doi:10.3390/ijerph20043130_

Round 1

Reviewer 1 Report

The topic addressed in the manuscript is interesting; however, the manuscript is missing a group of information for a more concrete analysis of the manuscript and better understanding of the results, namely.

1 - A map of the different karst types in the region, as karstic terrain is composed of various features and is a special aquifer.

2 - The map of relief features and water distribution.

3 - In reality besides the route of the roads and railroads the marginal uses were considered, but there is no map or image with the uses of the soil, etc...

4 - The indexes used for the different aspects deserve better explanations in the methods item.

5 - There are no images (photographs) of degraded areas at the different levels and of the original areas that exemplify the different levels of degradation.

On the other hand, all results relate aspects of uses (roads, agriculture, urbanization, tourism, etc.) with the loss of vegetation but not with one of the main characteristics of karst terrain, which is the dynamics of surface and subsurface waters, since they are aquifers with special characteristics. Desertification should strongly affect the water dynamics and the quality of the aquifer. The figures need improvement, such as increasing the font size and clearer legends. Tables 1 and 2 deserve more explanation in the text.

Reviewer 2 Report

Introduction

1. Add a brief introduction to the global karst background.

2.“Facing the ecological vulnerability and rocky desertification of karst, different experts and scholars have different understandings and definitions on its research and governance.” Why are all the opinions of Chinese experts and scholars listed?

Materials and Methods

1.Why do you choose Shilin Yi Autonomous County of Yunnan Province as the research object?

2.Point 2.3.1 Land-use classification,“the land in the study area is divided into 6 types, namely, rocky desertification free, potential rocky desertification, light rocky desertification, medium rocky desertification, severe rocky desertification and extremely severe rocky desertification.”,What are the rules for determining the degree of rocky desertification?

3.Point 2.3.3. Landscape index, add a brief introduction to landscape pattern index.

Conclusions

Add the significance of the research conclusions to the development of karst ecologically fragile areas and other regions of the world. The future direction of the research topic is also necessary.

References

Increase the proportion of English references, especially the latest papers.

Some more relative references should be added in the section of Introduction and Discussion.

The dominant driving factors of rocky desertification and their variations in typical mountainous karst areas of Southwest China in the context of global change

A novel remote sensing monitoring index of salinization based on three dimensional feature space model and its application in the Yellow River Delta

The Changes of Spatiotemporal Pattern of Rocky Desertification and Its Dominant Driving Factors in Typical Karst Mountainous Areas under the Background of Global Change

Round 2

Reviewer 1 Report

The modifications performed by the authors brought significant improvements to the manuscript. However, the figures still deserve improvement, as pointed out in the text. The inclusion of the data in Table 1 raises the question of whether it would be better to use the term degradation instead of desertification. I believe that the authors should ponder on this.
